# Toughening colloidal gels using rough building blocks

Florence J. Müller[1], Lucio Isa ®[1] & Jan Vermant ®[1] ✉

Colloidal gels, commonly used as mesoporous intermediates or functional materials, suffer from brittleness, often showing small yield strains on the order of 1% or less for gelled colloidal suspensions. The short-range adhesive forces in most such gels are central forces—combined with the smooth morphology of particles, the resistance to yielding and shear-induced restructuring is limited. In this study, we propose an innovative approach to improve colloidal gels by introducing surface roughness to the particles to change the yield strain, giving rise to non-central interactions. To elucidate the effects of particle roughness on gel properties, we prepared thermoreversible gels made from rough or smooth silica particles using a reliable click-like-chemistry-based surface grafting technique. Rheological and optical characterization revealed that rough particle gels exhibit enhanced toughness and self-healing properties. These remarkable properties can be utilized in various applications, such as xerogel fabrication and high-fidelity extrusion 3D-printing, as we demonstrate in this study.

Colloidal gels are a widely used class of soft materials that have a broad range of tuneable functionalities due to their mesoporous structure. Their technological applications range from food and pharmaceutical formulations—where their ability to impart gravitational stability is well known and exploited—to electrochemistry[1], direct ink writing in 3D-printing[2], tissue engineering and regenerative medicine[3] as well as emerging fields such as soft robotics[4]. The most commonly exploited functionality is that these materials can undergo a stress-induced, solid-to-liquid transition. The mechanical properties at rest arise because of the presence of a transient space-filling network, which then gets broken down and reorganizes when the material liquifies. However, the complex microstructure is also the reason for generally rather weak mechanical properties[5], a typical and very pronounced brittleness[6] and the occurrence of complex time-dependent behavior, known as thixotropy[7,8] which leads to flow-history-dependent microstructures and properties.

Much effort has been spent on understanding the link between the linear viscoelastic (LVE) response and the formulation parameters. The magnitude of the elastic modulus can easily be engineered to span several orders of magnitude[9], with key formulation variables being the particle volume fraction, size and shape, and the strength of the attractive interaction[10–14] as well as shear history[15]. Essentially, gels can

be viewed as disordered spring networks, with both the spring force and the network structure dependent on the colloidal interactions. The origin of the spring force has been suggested to have different origins, where clearly the adhesive nature of a short-range attraction between the particles plays a direct role[16]. Moreover, locally dense packing within gel strands can lead to particles in an isostatic condition that provides an elastic response[17]. The microstructure controls a dynamic localization length of the particles[18], as well as the number of interacting neighboring particles, with fractal structures dominating at low volume fractions[19] and a glassy structure emerging at high solid loading, which grants the application of mode coupling theory[20]. Thus, the role of structural heterogeneity is important, and small changes in microstructure can have dramatic effects on macroscopic properties[21,22].

Nevertheless, controlling and predicting the non-linear behavior of colloidal gels in terms of yielding and fracture is more complex: The macroscopically observed strains at which gels start to deform plastically can be very small compared to, e.g., amorphous glassy materials and is typically of the order of $10^{-2}$ or less[23]. Such small critical strains point to the presence of highly localized plastic deformations, strand plasticity or localized rupture of particle bonds which propagate through the system[24–27] rather than fully homogeneous, cooperative

[1]Department of Materials, ETH Zurich, Switzerland. ✉e-mail: jan.vermant@mat.ethz.ch

deformations. The corresponding brittle nature of most gels also limits the range of applications. When the material is deformed past the yielding transition, aggregates will break up and erode, and when vorticity is present in the flow field, aggregates will typically densify[28]. The consequent thixotropic response leads to a slow structural recovery, especially in dispersed systems, which puts an additional and severe constraint on the application domain of such materials.

In the present work, we remove these limitations by developing a chemistry agnostic method to increase the yield strain of colloidal gels and enable their rapid thixotropic recovery by engineering the particle surface topography. These novel building blocks enable control over gel elasticity in the usual manner, but the gels are tough, have flow-independent porosity and are essentially non-thixotropic, with fast and full recovery. Roughness has been shown to play a role in dense colloidally stable suspensions, which show discontinuous shear thickening as the particles get close[12,29–31]. In gels, particles are forced into contact, as evidenced by detailed studies of macroscopic aging[32]. This work investigates the addition of surface roughness to the building blocks, which can be expected to change the nature of the particle-particle interactions from being central (in the smooth case) to non-central (in the rough case), and can be used to engineer macroscopic rheological properties. The rationale is that rough particles can interlock with each other, while smooth particles can be expected to roll and slide when external shear forces are applied[33]. Yet, it also may extend the range of the interaction force, as it takes more deformation to disentangle rough particles from each other upon separation compared to smooth particles, similar to the mechanism of Velcro®.

Here, we designed our colloidal systems so the effects of surface roughness on the gel system can be studied independently, keeping all other parameters the same. As colloidal gels are strongly thixotropic, which makes their properties dependent on the mechanical history, even loading the sample into a measurement device can strongly modify the structure. Commonly, pre-shear protocols are used to condition the samples[34]; however, surface roughness of the primary particles is expected to influence the structure under flow and hence using a pre-shear protocol would bias the measurements. This work therefore details an approach to exclusively compare the influence of primary particle surface roughness on the macroscopic gel properties under shear conditions. We use a thermoreversible system, where the gel can be formed inside the measurement cell, by

changing the temperature, in a similar way for both smooth and rough systems.

Thermoreversible gels often consist of core-shell particles, where the interaction of the shell with the suspending media changes with temperature. Here, silica ($SiO_2$) core particles were chosen because silane chemistry allows for the engineering of different particle morphologies and surface structures[35]. An octadecyl ($C_{18}$) brush is used as a thermoresponsive grafting agent, which undergoes a change in interaction with tetradecane ($C_{14}$) as a suspending media[36,37]. For higher temperatures, the sample is liquid: the octadecyl brush is dis-ordered and solvated in tetradecane and extends into the bulk phase, thus forming a steric hindrance layer. At lower temperatures, the octadecyl undergoes a phase transition to an ordered crystalline state, inducing a density change that increases van der Waals attractions, which makes the particles assemble into a spanning network structure that forms a gel (see Fig. 1b). This effect is reversible, meaning that upon heating of the sample, the gel will fluidify.

The thermoresponsive $SiO_2$ core and octadecyl brush model system was developed by Van Helden et al.[38,39] in 1980 and since then used as a model system in many different suspending media such as decalin[14,40], tetradecane[13,40,41], dodecane[42] and hexadecane[43–45]. Van Helden reports the direct esterification of the octadecanol to the surface groups of the $SiO_2$ particles. This reaction was found to be difficult to control in terms of grafting density which also yields different liquid-to-gel transition temperatures.

This work reports the development of a highly robust and versatile two-step grafting method shown schematically in Supplementary Fig. S1. It is based on a secondary amine-yne click-like-reaction, previously reported to crosslink hydrogels[46], and has been adapted to graft $SiO_2$ particles (Fig. 1a). We then investigate the rheological and structural properties of gels of these particles as a function of temperature, volume fraction, and surface roughness, demonstrating that the latter property offers exciting possibilities to modify the shear response of colloidal gels for processing.

## Results and discussion

We developed a two-step grafting method to reproducibly bind octadecyl to the surface of rough and smooth $SiO_2$ particles at comparable coverages (Fig. 1a, "Methods," Supplementary Information 1, Supplementary Fig. S1). In the quiescent state, the network is expected to be similar for smooth and rough primary particle systems because the

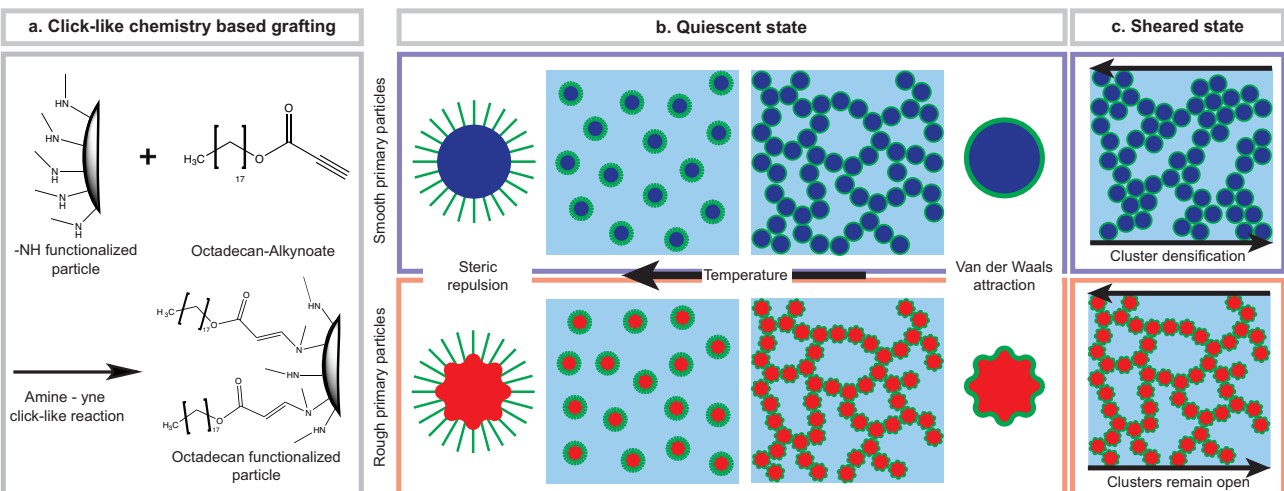

**Fig. 1 | Synthesis approach and gelation mechanism of our thermoresponsive colloidal gel with smooth and rough primary particles. a** Grafting approach using amine-yne click-like-chemistry, where smooth or rough -NH functionalized particles are grafted with alkynoate functionalized octadecane. **b** Scheme of the quiescent gelation mechanism, where both smooth (blue) and rough (red) particles have a steric repulsion at high temperatures and network-forming phobic interactions at low temperatures. **c** Scheme of the smooth (blue) and rough (red) primary particle gel network during and after shearing.

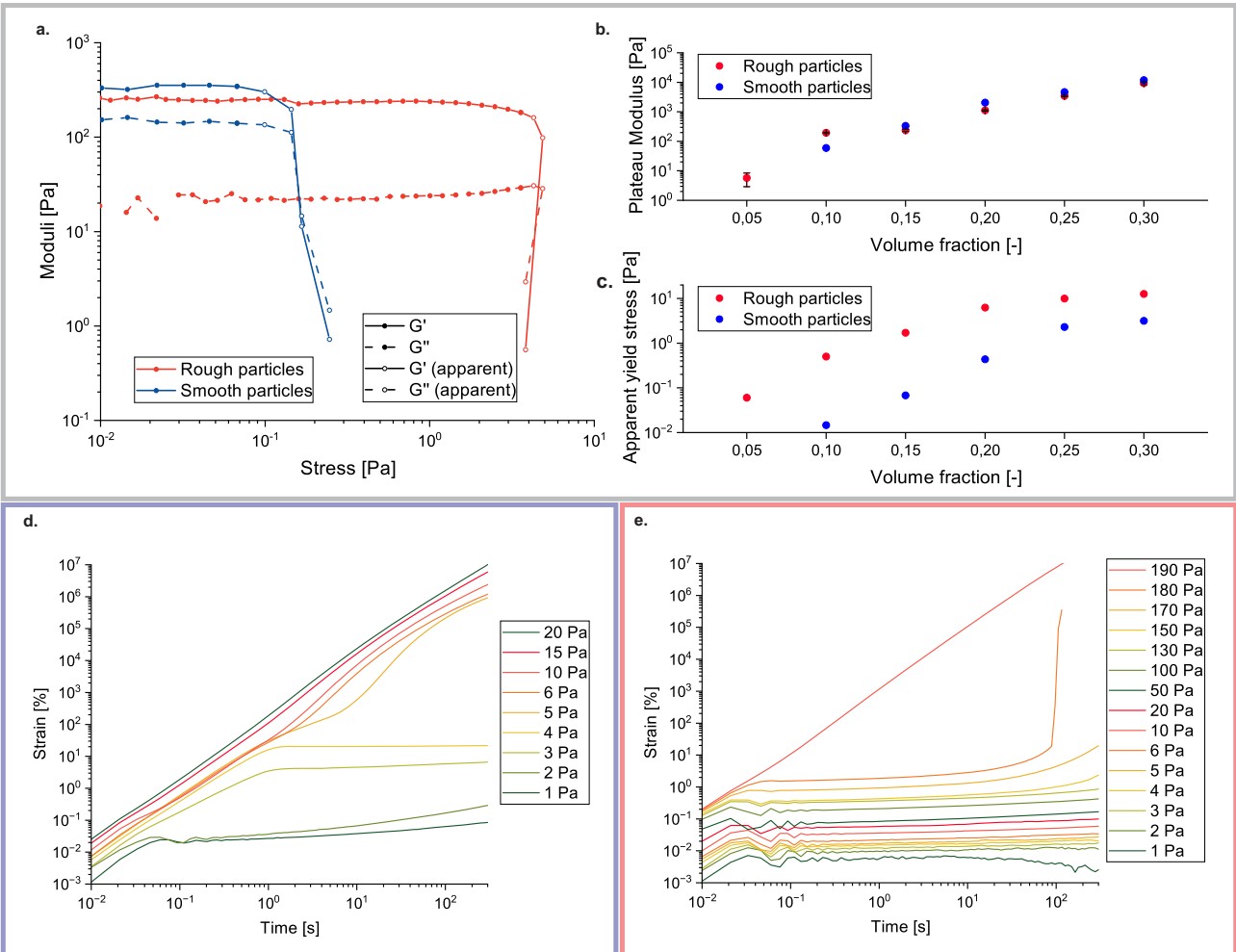

**Fig. 2 | Rheological characterization of the yielding of smooth and rough particle gels. a** Example stress amplitude sweep (elastic modulus G' and loss modulus G") of a smooth (blue) and rough (red) primary particle gel at equal volume fraction ($\phi = 0.15$), where the solid symbols represent reliable data and the empty symbols represent data points that might be affected by measurement errors (see text). **b** Elastic plateau moduli (G') for smooth (blue) and rough (red) particle gels of different volume fractions. **c** Apparent yield stress (last data point in LVE) for smooth (blue) and rough (red) particle gels for different volume fractions. **d** Creep measurements for a smooth particle gel at $\phi = 0.25$ (G'$_{Plateau}$ = 4615 Pa). **e** Creep measurements for a rough particle gel at $\phi = 0.25$ (G'$_{Plateau}$ = 3414 Pa).

same mechanism induces attractive interactions. However, during and after deformation, we expect the network geometry to change due to surface roughness (Fig. 1b). The rough particles have the ability to interlock, thus resisting shear-induced densification, while the smooth particles can roll against each other and densify into clusters, which are then more difficult to break up (Fig. 1c). We hence expect differences in the yielding transition and in the thixotropic response.

The difference in the yielding behavior for rough (RMS = 7.2 mm) and smooth primary (RMS = 0.7 mm) particle gels can be detected through rheological characterization. Figure 2a shows an example of a stress amplitude sweep for $\phi = 0.15$ gels of rough and smooth particles. The elastic moduli ($G'$) of both suspensions are comparable in magnitude, while the onset of the yielding transition, in terms of the shear stress amplitude where the modulus drops, is delayed by more than an order of magnitude for rough primary particle gels. For the gel made up of smooth particles, the transition takes place at 0.1 Pa ($\gamma = 0.018\%$), whereas for the gel made up of rough particles, this shifts to 20 Pa ($\gamma = 1.9\%$). Through the shape of the amplitude sweeps, where the moduli drop very suddenly for increasing shear stresses, the very abrupt transition of the solid to the liquid state is clear - to a point where it is actually difficult to resolve it by the instrument, as the stress amplitude applied by the instrument decreases. Empty symbols indicate data potentially affected by the instrument feedback loop being

too slow or the occurrence of edge fracture effects. This is why the data points were conservatively interpreted by marking the points that could be trusted using a solid symbol. Further stress amplitude sweeps for all volume fractions are given in Supplementary Fig. S3a, b, and the apparent yield strain for all volume fractions is represented in Supplementary Fig. S3c.

Figure 2b shows the elastic modulus $G'$ over the volume fraction $\phi$ for both particle types. Surprisingly, the evolution of $G'$ with volume fraction is the same for smooth and rough particles, except for the lowest volume fractions, where a somewhat lower percolation threshold is observed for the rough particles ($\phi = 0.05$). In a sparsely populated system close to the percolation threshold, the additional surface area of the rough particles allows for stronger network formation, while at higher volume fractions ($\phi > 0.1$), the system is more crowded, meaning that there is no need for additional surface area to contribute to network formation.

When considering the apparent oscillatory yield stress (Fig. 2c), which we define here as the last point in the LVE region, i.e., corresponding to a >10% difference in G' with respect to the average of the previous points (sometimes called the perturbative yield strain[23]), the rough particle systems yield at significantly higher stress amplitudes compared to the smooth particle systems at the same volume fraction. This delay in yielding is even more clearly observable in continuous

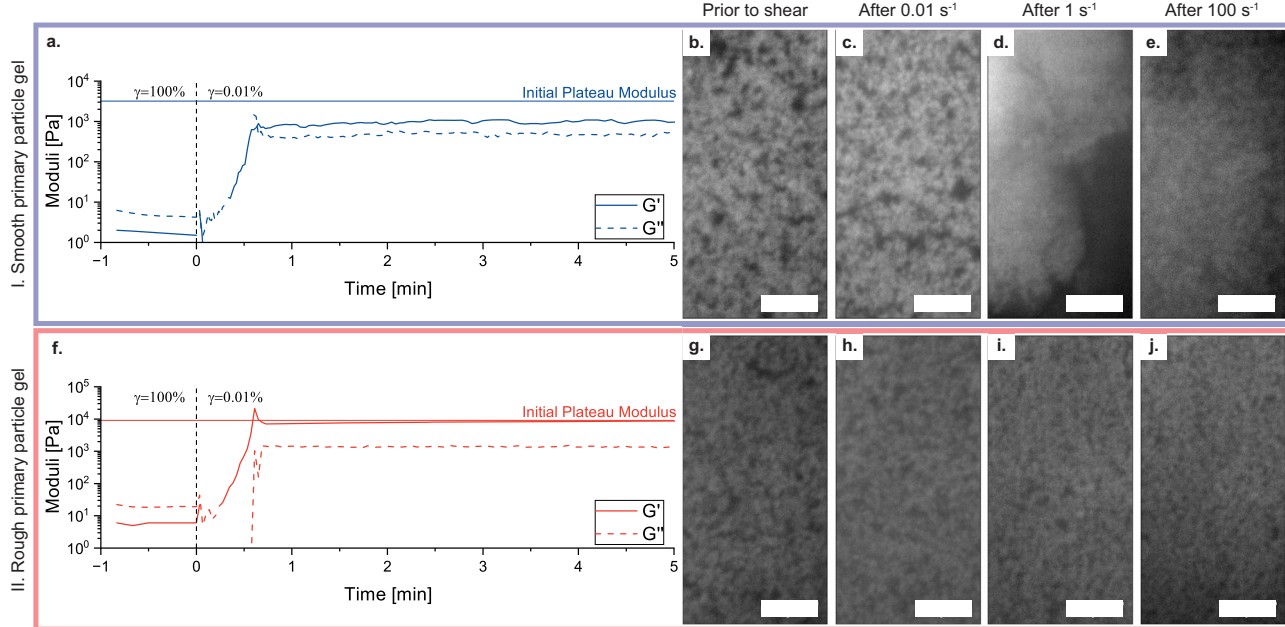

**Fig. 3 | Recovery characteristics of (I.) smooth and (II.) rough primary particle gels. a** Shear recovery measurement of a $\phi = 0.25$ smooth particle gel in the LVE region after fluidification at 100% oscillatory strain amplitude $\gamma$ with the elastic modulus G' and the loss modulus G". **b–e** Microstructure of smooth particle gel **b** after quiescent gelation, **c** after shearing at 0.01 s$^{-1}$, **d** after shearing at 1 s$^{-1}$, **e** after

shearing at 100 s$^{-1}$. **f** Shear recovery measurement of a $\phi = 0.25$ rough particle gel in the LVE region after fluidification at 100% oscillatory strain amplitude $\gamma$ with the elastic modulus G' and the loss modulus G". **g–j** Microstructure of rough particle gel **g** after quiescent gelation, **h** after shearing at 0.01 s$^{-1}$, **i** after shearing at 1 s$^{-1}$, **j** after shearing at 100 s$^{-1}$. All scale bars are 5 μm.

creep experiments, represented in Fig. 2d, e for the smooth and rough particle cases, respectively. For smooth particle gels at a volume fraction of $\phi = 0.25$, an elastic regime is observable for low stresses (<4 Pa) with even some creep ringing at short time scales. For the time scales accessed in this experiment, the material then starts to yield gradually (over the time scale probed) at a stress of 5 Pa to be fully fluidified at 15 Pa, which corresponds to 0.3% of the plateau modulus. For the rough particle gel at the same volume fraction of $\phi = 0.25$ (Fig. 2e), the elastic regime, with more pronounced creep ringing extends until 130 Pa. The rough particle gel starts to fluidify when it is sheared at 180 Pa for 100 s (corresponding to a strain of 18.9 and 5.3% of the plateau modulus), showing a very sudden and dramatic increase in strain[47] which was also observed in the oscillatory measurements described above. We hypothesize that this sudden fluidification is due to a general breaking of many interparticle bonds at the same time for the rough particles, whereas smooth particles tend to rearrange as they are sheared.

Delayed and more abrupt yielding is not the only characteristic that differentiates rough and smooth particle gels. Although the particles were selected for their rheological contrast and not optimized for confocal imaging in terms of size and refractive index matching, Fig. 3 still reveals the consequence of interlocking of the rough particle asperities through oscillatory recovery experiments and confocal microscopy after continuous shearing. The rheological data in Fig. 3a, b show an oscillatory recovery measurement, where smooth and rough primary particle gels at a volume fraction of $\phi = 0.25$ are first sheared at $\gamma = 100\%$, which fluidifies both gels (smooth gel $\sigma_{max} = 7.4$ Pa, rough gel $\sigma_{max} = 25.9$ Pa). Then, they are left to recover in the LVE region at $\gamma = 0.01\%$ (smooth gel $\sigma_{equ} = 0.23$ Pa, rough gel $\sigma_{equ} = 0.98$ Pa). The smooth particle gel never recovers to the initial plateau modulus (indicated by the continuous horizontal line), even for much longer times than 5 min. For the rough primary particle system, a full recovery of the elastic modulus occurs after 3.5 min, and the measurement is stable compared to the smooth one. The reason for the full and quick recovery in the rough particle gel is the fact that particles can interlock,

which prevents the particles from densifying into compact clusters. The fluidification of rough gels results in the breakup of open flocs, which can recover their initial geometry more easily. The dense clusters that occur during the shearing of smooth particle gels are not able to separate when shear subsides, and thus the gel will reform with much larger and more compact primary clusters. This is also apparent in the confocal images in Fig. 3, where the structure prior to shear is compared to the structures after 0.01 s$^{-1}$, 1 s$^{-1}$ and 100 s$^{-1}$ of shearing (intermediate shear rates as well as the Moran's Index analysis of the images are shown in Supplementary Fig. S5). The structures prior to shear of the rough and smooth particle gels are quite comparable. In the confocal images of the smooth particle gel (Fig. 3a), there are some cracks and larger voids forming after shearing at 0.01 s$^{-1}$. After shearing at 1 s$^{-1}$, there is a clear particle-rich and particle-poor region in the image. When sheared at 100 s$^{-1}$, the sample shows a particle-dense region without an apparent network structure; the particles have been broken apart by the shear forces, but some clusters seem to remain. In the confocal images of the rough particle gel (Fig. 3g–j), the network structure prevails throughout all images, indicating that interlocking of the surface asperities might hinder flow densification.

The rheological and optical characterization both show that, while being comparable in the quiescent state, smooth and rough particle gels differ during and after yielding. In smooth gels, the plastic events and particle bonds rupture very locally and occur progressively. The rough particle gels, similar to Velcro®, pull apart more slowly and are more likely to reattach due to the non-central forces and interlocking mechanisms. Therefore, the deformation is homogeneous, but the final structure collapses suddenly. This correlation is further highlighted by studying systems of particles with asperities of different sizes relative to the same 750 nm core. The difference can be rationalized by a roughness factor, defined below:

$$\text{Roughness factor} = 100 \, \frac{\text{asperity size}}{\text{core size}}. \tag{1}$$

**Table 1 | Critical strain from the creep experiments as a function of roughness for particles with a 250 and 750 nm core diameter**

| Roughness factor | 0 | 10 | 0 | 2 | 4 | 8 |
|---|---|---|---|---|---|---|
| Core size [nm] | 250 | 250 | 750 | 750 | 750 | 750 |
| Asperity size [nm] | 0 | 25 | 0 | 15 | 30 | 60 |
| Critical strain [%] | 0.3 | 18.9 | 0.19 | 0.44 | 5.23 | 10.2 |

Table 1 shows the values of the critical strain for particles of different roughness factors (core size 750 nm) in addition to the previously shown system of particles (core size 250 nm). The critical strain at which the structure starts to yield in a continuous flow condition increases with increasing roughness factor. This is in line with the previous argument, where for rough particles, the interlocking asperities detach at larger strains and can rearrange due to non-central forces. This effect scales with asperity size because the larger the asperity, the more strain is needed to pull two particles apart for a given core size. Here, we note that the thermoresponsive coating (octadecyl) is much smaller in length than the imparted asperities for the roughness factors 4, 8 and 10. The octadecyl brush has a length of 2.6 nm at 5.5 °C and 3 nm in the swollen case, as measured with AFM.

The reversible solid-to-liquid transition and the sensitivity to deformation histories are key during many manufacturing processes, where the behavior and properties during flow and their recovery thereafter are crucial. Here, we investigate the influence of surface roughness on extrusion 3D-printing and xerogel fabrication of colloidal gels.

In extrusion 3D-printing, shear deformations are important, hence shear-thinning and self-healing properties are crucial in order to achieve high printing fidelity. Shear-thinning materials are injectable and can be extruded through a nozzle, and a self-healing gel can reform upon deposition on the substrate. Figure 4I shows extrusion 3D-printed lattice structures using smooth (blue, Fig. 4a–d) and rough (red, Fig. 4e–h) primary particle gels with $\phi = 0.25$ under the same printing conditions. It is apparent from the images that the printing quality of the rough particle gel is superior to the smooth particle gel, as the extruded material across the different layers clearly retains a more defined shape for the rough particle system. This is primarily because of the previously described recovery ability of the rough particle. Macroscopically, this results in a precise deposition of a filament that stays stable over time and a structure that does not slump. In the structure deposited with the smooth particle gel, slumping is observable shortly after printing, in addition to a phase separation of the solid and liquid phase induced by the densification of the smooth primary particles during the extrusion process.

Figure 4II compares the suitability of gels made of smooth and rough particle gels to create mesoporous xerogels through supercritical drying. The difficulty in making these materials is the assembly of a 3D-spanning and interconnected network over several length scales[48]. Here, surface roughness plays an important role in keeping the network connected, even during the drying process, through mechanical interlocking of the particle asperities. Figure 4k shows images of the macroscopic gels made out of smooth (left) and rough (right) primary particles through the same process. Before the supercritical drying step, both gels were cylindrical; however, the rough particle gel kept its cylindrical shape, and the smooth particle gel slumped and collapsed into an amorphous shape. The slumping and densification of the smooth particle gel also become apparent when comparing the SEM images of the smooth (Fig. 4i, j) and rough (Fig. 4m, n) particle gels, where the smooth particle gel is very dense and the network structure is barely detectable in the image. The microstructure of the rough particle gel is significantly more open and

shows pores on a microscopic scale. This was also detected in the Brunauer-Emmett-Teller (BET) measurements, where the surface area was measured and normalized by the weight of the sample. Figure 4l shows the BET measurements for the smooth and rough particle powders and gels. In both smooth and rough cases, the surface area increases when a gel is formed, however, the increase in surface area is higher for the rough gel. It should be pointed out that the particle sizes here are not optimized for gel performance, but this experiment provides a comparison of the effects of roughness on the openness of the microstructure. The insensitivity of the rough gel system to flow history opens up the possibility of extruding these into desired shapes and then drying them supercritically, providing a path for novel types of extrudable xerogels and aerogels. This would require smaller particle sizes, but the concepts remain the same.

In summary, this work details how colloidal gels can be rendered tougher using rough building blocks. We report on a robust and reproducible core-shell colloidal gel model system with thermoreversible features that allow for a non-biased comparison of particles with different surface roughnesses. The silica core particles were functionalized with octadecyl using a highly efficient click-like-chemistry approach. Rheological characterization showed that imparting surface roughness to the primary particles increased the yield stress by an order of magnitude and the yield strain by as much as two orders of magnitude while keeping the moduli the same. Furthermore, the rough particle gel showed a self-healing ability, whereas the smooth particle gel did not. The underlying mechanism is explained through the interlocking of the rough particle asperities, meaning that larger strains are needed to detach a connection. Additionally, surface roughness allows the particles to reattach more easily. When a certain critical strain is reached, the structure will fluidify suddenly, which is represented in the creep behavior. The self-healing properties were explored in possible applications, such as xerogel fabrication and extrusion 3D-printing, where the rough particle systems enabled more precise structures. Surface roughness in colloidal systems can be highly relevant for different industrial applications where they are subjected to flow. In conclusion, rough particle systems that are able to keep an open network structure during flow and rebuild the structure upon flow cessation circumvent the pitfalls of current colloidal gels in a simple manner and can be used to intrinsically design formulations with a range of properties without the need of rheological additives.

## Methods
### Materials
Ethanol (99.8%, Merck), Tetraethyl orthosilicate (TEOS, 99%, Sigma-Aldrich), Ammonia solution (NH₄OH, 25%, Merck), Poly-diallyldimethylammonium chloride (Poly-DADMAC, 400–500 kDa, 20 wt%, Sigma-Aldrich), Trimethoxy[3-(methylamino)propyl]silane (MAPTMS, 97%, Sigma-Aldrich), 1-Octadecanol (Octadecanol, 99%, Sigma-Aldrich), Toluene (99.85%, Fisher Scientific), Toluenesulfonic acid-p monohydrate (pTsOH, 98%, Sigma-Aldrich), Propiolic acid (96%, Sigma-Aldrich), Isopropanol (technical grade), Deionized water (Milli-Q, Merck-Millipore). All products were used as received.

### Particle synthesis
**Smooth and rough core particle gel synthesis.** Silica particles (300 nm diameter) were synthesized using the Stöber process. In a typical synthesis, 200 ml of ethanol, 18 ml of MiliQ and 10 ml of ammonia were stirred in a 500 ml glass bottle at 500 rpm. Then, 12.4 ml of TEOS was added quickly to the solution, which was then left to react for 24 h. Next, 3.1 ml of MAPTMS solution (5 v% in ethanol) was added to the suspension (without a cleaning step) using a syringe pump (2ml/h) and left to react for 1 h after the end of the injection. The particles were then cleaned 1x with ethanol and 2x with isopropanol with centrifugation and redispersion steps.

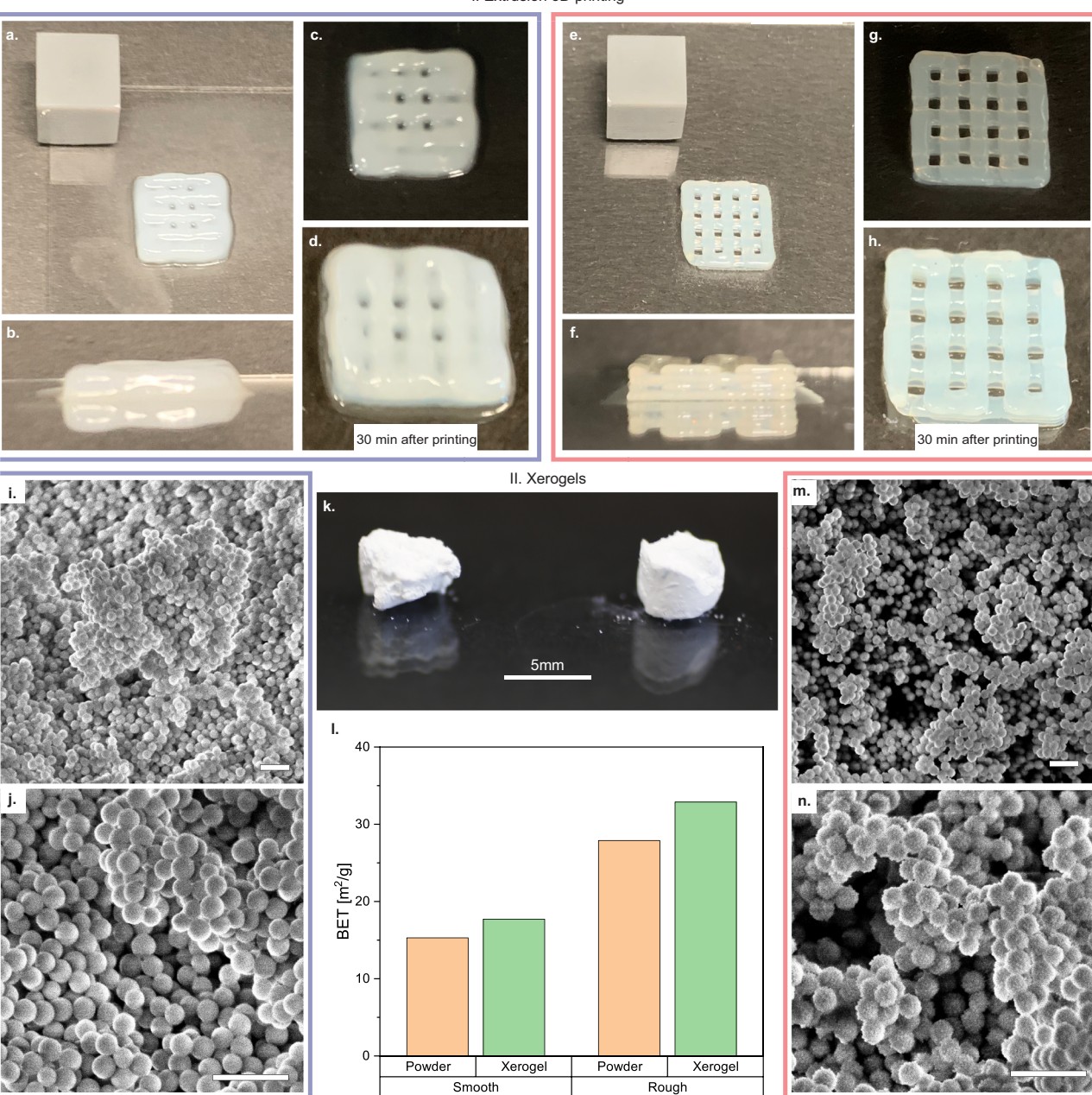

**Fig. 4 | Applications of rough particle systems.** I. Extrusion 3D-printing (**a**–**h**) **a**–**c** Printed 4-layer lattice structure with a smooth particle gel (cube is 1 × 1 × 1 cm), **d** same smooth particle gel structure after 30 min of aging, **e**–**g** printed 4-layer lattice structure with a rough particle gel (cube is 1 × 1 × 1 cm), **h** same rough particle structure after 30 min of aging. II. Xerogels (**i**–**n**) **i**, **j** SEM image of smooth particle gel (scale bar 1 μm), **k** image of the smooth particle gel on the left and rough particle gel on the right, **l** BET measurement for smooth and rough particle powder and gel, **m**, **n** SEM image of rough particle gel (scale bar 1 μm).

Silica particles (250 nm diameter) were synthesized by the Stöber process. In a typical synthesis, 200 ml of ethanol, 18 ml of MiliQ and 8 ml of ammonia were mixed in a 500 ml glass bottle at 500 rpm. Then, 12.4 ml of TEOS was added quickly to the solution, which was then left to react for 24 h. The particles were cleaned 2x with ethanol and 3x with MiliQ. A mass of 1 g of particles was then suspended in 300 ml of MiliQ with 0.068 ml of Poly-DADMAC and stirred for 1 h. The positively charged particles were then cleaned from excess Poly-DADMAC in three centrifugation and redispersion steps with MiliQ. The particles were then dispersed in 200 ml of MiliQ with 500 μl of Ludox particles (LM 50) and left to stir for 40 min in order for the berry particles to adsorb to the positive cores. The particles were then cleaned with water 2x, and the solvent was exchanged for ethanol. The rough

particle assemblies were solidified and functionalized by the addition of 3.1 ml of MAPTMS solution (5 v% in ethanol) using a syringe pump (2ml/h)—the reaction was left to proceed for 1 h after the end of the injection. The particles were then cleaned 2x with ethanol and 2x with isopropanol with centrifugation and redispersion steps.

**Octadecane-alkynoate functionalization.** The octadecanol was functionalized with an alkynoate group, following the Fischer esterification process. For this, 10 g (1 eq) of octadecanol was dissolved in 70 ml of toluene at 50 °C in a 100 ml roundbottom flask. After complete dissolution, 0.5 g (5% of octadecanol weight) of pTsOH was added to the reaction, followed by 2.52 ml (1.1 eq) of propiolic acid. A Dean-Stark trap and a condenser were then installed on the roundbottom flask, and the

solution was heated to 135 °C to start the reaction, which was left for 24 h. The octadecane-alkynoate was purified by evaporating the toluene; the product was then dissolved in 20 ml of acetone and dropped into iced MiliQ for the octadecane-alkynoate to precipitate. The suspension was then isolated using a vacuum filter and dried under vacuum at 30 °C for 24 h. The product was stored in the freezer to avoid degradation.

**Amine-yne click-like-reaction.** The octadecane-alkynoate was dissolved in isopropanol at 5 wt% at 40 °C. To graft 1 g of secondary amine functionalized core particles (smooth or rough), 1.25 ml of the octadecane-alkynoate stock solution was added to a 10 wt% suspension of particles in isopropanol at 40 °C and left to stir for 3 h. The functionalized particles were then washed with isopropanol three times, dried in a rotary evaporator and subsequently in a vacuum oven for 48 h.

**Gel formation.** The dry octadecyl functionalized particles were suspended in tetradecane and heated to 60 °C for 20 min in a water bath. In order to ensure the complete dispersion of the particles, the suspension was tip-sonicated for 10 s to avoid aggregates.

### Characterization methods
**Particle characterization.** The size and shape of the particles were assessed using SEM imaging (REM-LEO1530, Zeiss, Germany) and AFM (Dimension icon, Bruker).

**Rheology measurements.** Rheology measurements were performed on an Anton Paar MCR 502 using a 20 mm plate-plate geometry with a pillar roughness of 100 μm on the top and bottom. The sample was loaded at 60 °C, cooled down to 5 °C and equilibrated for 30 min. Between the measurements, the gel was rejuvenated by cycling the temperature to 60 °C for 20 min and cooled to 5 °C for 30 min.

All measurements were performed at 5 °C. The stress amplitude sweep measurements were performed at 1 rad/s. The continuous creep measurements were performed for 5 min. In the recovery measurements, the sample was sheared at 100% strain and 1 rad/s for 1 min before letting the structure recover at a strain of 0.01%.

**Rheoconfocal measurements.** Rheoconfocal measurements were performed using a previously developed custom setup[49] where the lower plate consists of a glass slide, through which a confocal microscope scans through the sample in z-direction. The temperature was controlled using a commercially available Peltier hood (Peltier Temperature Device 200, Anton Paar) through a heated or cooled nitrogen stream on the upper geometry (20 mm with a pillar roughness of 100 μm). The glass slide that served as lower geometry plate was functionalized with octadecyl in order to reduce slip during the measurement. The startup flow measurement was performed at the flow rates of 0.01, 0.1, 1, 10, 100 1/s for 1000% strain.

### Extrusion 3D-printing
The printed structures were fabricated by loading a $\phi = 0.25$ weight fraction gel (smooth and rough primary particles) in tetradecane (0.8 ml of sample) into a 1 ml syringe before heating the gel to 60 °C to eliminate shear history. The syringe was then fitted with an 18 gauge extrusion tip (1.27 mm). The printed structure was a $1 \times 1$ cm square with three intermediate lines spaced 2 mm apart. This layer was rotated 90 °C for every even layer, and a total of four layers was deposited at 1 mm/s. The print was performed at room temperature, and the glass slide substrate for the print was placed on a metal plate cooled to 5 °C. The printed samples were kept at 5 °C to age.

### Xerogel fabrication
Xerogels were made by forming a $\phi = 0.25$ weight fraction gel (smooth and rough primary particles) in tetradecane and filling it into a 1 ml

syringe with the top cut off. The gel was then left to age at 5 °C for several hours and transferred into a sieve, submerged in acetone to exchange the solvent overnight. The sieve with the sample was loaded into a critical point dryer (SPI supplies, 13200-AB) at 10 °C and flushed with $CO_2$ in three cycles (5 times each) with a 15-min soaking time. After the acetone was evacuated from the system, the temperature was increased to 42 °C to achieve a supercritical state of $CO_2$ before venting the dryer over 30 min. The powder and xerogel surface area was assessed through $N_2$ gas adsorption measurements on a Quantachrome Autosorb iQ at 77 K. Prior to the measurements, the samples were outgassed for 24 h. The surface area was evaluated by the Brunauer-Emmett-Teller (BET) method, and the pore size distribution was determined by a density functional theory (DFT) analysis using a Non-Local DFT (NLDFT) calculation model for nitrogen at 77 K on cylindrical pores in silica.

## Data availability
The authors declare that the data supporting the findings of this study are available within the paper and its supplementary information files. Source data are provided with this paper.

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

## Acknowledgements

The authors acknowledge funding from the Swiss National Science Foundation, project number 200020-192336 as well as the International Fine Particle Research Institute (IFPRI). We acknowledge Elena Tervoort-Gorokhova and David Kiwic for their input and help on xerogel fabrica-tion as well as Shivaprakash N. Ramakrishna for his help with supporting AFM experiments. We also thank Madhu V. Majji, Theo Tervoort, Stephan Busato, Vincent Niggel, Pierre Lehéricey and Kirill O.J. Feldman for helpful discussions.

## Author contributions

J.V. and L.I. conceptualized the study; F.J.M. developed the model sys-tem, synthesized the samples, performed the experiments and post processed the data; J.V. wrote the original introduction; F.J.M. wrote the original results and methods section; J.V. and L.I. reviewed and edited the final manuscript.

## Competing interests

The authors declare no competing interests.
