## [Peer Review File · Nature Communications]

REVIEWER COMMENTS

Reviewer #1 (Remarks to the Author):

This is a very nice paper showing unambiguously that colloidal gels made of rough particles have significantly superior non-linear mechanical properties as compared to conventional gels made of “smooth” particles. This is something that people in the field knew (and often claimed) “intuitively”, at the qualitative level. The great contribution of this manuscript is to show it in a clean, quantitative way, using exactly the same kind of particles and modifying their roughness in a well-controlled way.

As such, the manuscript goes well beyond the existing literature and will certainly have a significant impact. Furthermore, the authors present two examples of applications where the enhanced mechanical properties of their gels could be key: extrusion printing and the fabrication of xerogels. For all these reasons, I strongly support publication in Nat. Comm.

I list below (in no particular order) several points I’d like the authors to address before final acceptance.

1) No real particle is actually “smooth”. Did the authors characterize the surface of the “smooth” silica particles? If not, they should cite previous works where this characterization was done for similar particles.

2) The authors claim that the yield strain of colloidal gels is typically $< 1\%$ (abstract and line 45). I found this value quite small: some gels do have yield strains as large as 30% or more (see e.g. T. Gisler and D. A. Weitz, Strain Hardening of Fractal Colloidal Gels, Phys. Rev. Lett. 82, 1064 (1999)). Please revise this statement.

3) Figure 1c: I find this sketch not very clear, one doesn’t see very well the cluster densification

4) In all figures: increase the font size of the axis labels

5) Figures 2d, 2e of the main text and Figs. 3a, 3b of the SI: I suggest using the same color coding as in the rest of the paper, e.g. put each graph in a box, blue or red for smooth and rough particles, respectively.

6) Figures 2d, 2e: I suggest adding to the data label the % of the applied stress as compared to the plateau modulus (and quoting the value of the plateau modulus for each gel in the caption).

7) Line 119: Fig. S1 3 shows stress sweeps, not strain sweeps.

8) Line 129: I suppose that the last point in the LVE is defined as the last point before G' drops by 10% or more upon further increasing the stress (10% with respect to what? The value of the previous point? The average of all previous points?). Please consider rephrasing this sentence to make it clearer.

9) The abstract and introduction put emphasis on the small yield strain that is typically found in colloidal gels, while the paper discusses almost only yield stress values. I understand that a stress-controlled rheometer was used, but it would be interesting to show (and discuss) at least the strain at yielding, e.g. as an additional panel after Fig. 2c of the main text. In the current version, strain at yielding is only briefly mentioned when discussing Table II.

10) Table II: it would be useful to mention also the thickness of the stabilizing layer, both at low and high temperatures.

11) Figures 3b-e and 3g-j: it would be nice to add some quantitative characterization of the length scale of the heterogeneities seen in the images, rather than discussing them just at the qualitative level. Perhaps a straightforward Fourier transform of the images could be informative, could the authors give it a try?

12) Please define the acronym BET, and provide details on the BET measurements in the Methods section.

13) Consider renaming the label "II. Aerogels" of Fig. 4 to "II. Xerogels" for consistency with the rest of the text and figure captions.

14) I'm puzzled by the modest increase of the surface/mass ratio in gels as compared to powders, for both smooth and rough particles (Fig. 4I). Did the authors try to produce xerogels starting with gels at ϕ lower than 25%? I understand that the part on xerogels is, at this stage, just a proof-of-

concept and that improving the BET ratio significantly would probably require a lot of work. Nevertheless, it would be nice to show data where the xerogel has properties that differ more markedly from those of powders, and more markedly between smooth and rough particles (the relative increment of the BET signal from the powder to the gel is in fact very similar for both kinds of particles)

15) Ref 29 needs to be fixed

Reviewer #2 (Remarks to the Author):

The authors create thermo-reversible particle based gels and perform experiments to distinguish their behavior from smooth particles. I do find the work to be compelling, interesting and novel, and worthy of publication, but I require the authors to make some major/minor improvements.

Major comments:

- The data shown in the main text lacks any form of statistical analysis. For example, in table 1, where is the average critical yield strain? Simply put. I am sure the authors performed many, if not hundreds of experiments, and yet none of this averaging is present in the manuscript. The reader is left to conclude that individual experiments are performed. The authors are required to add some statistical analysis.

- Throughout the manuscript, the authors discuss the individual particle behavior and an 'interlocking'. While this is likely true, there is no evidence in this paper to confirm this. The authors should either indicate that this is logical speculation, or show clear individual particle analyses to confirm their hypothesis - it is a hypothesis. Particle based simulations would be one logical route to gain this level of detail or in-situ shear confocal microscopy. Both however are likely beyond the scope of this work.

Minor Comments:

Page 3: by far the most common colloidal gels are made with smoothly droplets, i.e. emulsion. why has a discussion of these gels been ignored here?

Page 3: again, why no statistics on yielding? A thermos-reversible system allows for this quite easily.

Page 6: The authors state: “In a sparsely populated system, where particles have space to rearrange, surface roughness may bring additional stability for network formation, while at higher volume fractions ($\phi > 0.1$), the system is more crowded, meaning that there is no need for an additional interlocking mechanism to contribute to network formation.” I find this statement to be unsupported by the evidence. It is quite a bold claim and given the highest volume fraction of 0.30 which is not that 'crowded' there is still ample local free volume (nearest neighborhood) for particles to rearrange via rolling/sliding. The authors should restate or better verify this statement.

Page 6: The authors state “The rough particle gel starts to fluidify when it is sheared at 180 Pa for 100 s (corresponding to a strain of 18.9 %), showing a very sudden and dramatic increase in strain which was also observed in the oscillatory measurements described above. “ this creep data is quite striking (and also lacking in statistics) and should be discussed more than this limited discussion. For instance, is there an origin to the abrupt yielding behavior in the rough particles? Were proper yielding statistics performed? How does this compare with past creep yield experiment on colloidal gels, in particular J. Sprakel, et. al. PRL 2011?

Page 7 : The authors state “The smooth particle gel never recovers to the initial plateau modulus (indicated by the continuous horizontal line), even for much longer times than 5 min.” What is meant by 'initial plateau modulus'? Is this the modulus formed after the dispersion has been cooled in the rheometer? The shear history here is very important to mention compared to typical non-thermoreversible gels which have inherent loading history. This strengthens the authors' conclusions.

Page 7: the authors state “The dense clusters that occur during the shearing of smooth particle gels are not able to separate when shear subsides, and thus the gel will reform with much larger and more compact primary clusters.” While this statement is probably correct, the confocal images do not prove this. When considering, panel b compared to e, it is very difficult to conclude that denser clusters are formed. No statistical analysis of the structure factor is performed and should be. Indeed, the best and most reliable measurement would be SAXS determination, but I do understand that this is too much experimentation. I suggest the author either perform some analyses of their confocal images, or tone down the strength of their statement as it is currently speculation and not obvious from the images provided - images being very subjective and also highly subjected to the bias of the writer.

Page 7: the authors state “In the confocal images of the rough particle gel (Figure 3 b.), the network structure prevails throughout all images, showing that interlocking of the surface asperities hinders flow densification.” This conclusion is again speculation and should be mentioned as such. And the figure reference is incorrect, it should not be 'b'.

Page 8: the authors state “It is apparent from the images, that the printing quality of the rough particle gel is superior to the smooth particle gel, primarily because of the previously described recovery ability of the rough particle.” What is apparent? Please describe better the print quality. This sentence is too vague and non-descriptive. Additionally, the recovery is 0.5minutes for the rough particle gel, as shown in the above figure. This is still extremely slow recovery compared to millisecond for polymeric 3D printing. The authors must discuss the printing velocity used, and how this affected the print quality. The methods section is severely lacking in detail for this application.

Conclusion: Again there is strong speculation that ‘interlocking’ is the origin of the rheological difference, but no local individual particle data is provided in this manuscript. I do not believe the absence is too impactful of the results, rather than the authors temper their language to indicate that it is hypothesized and not confirmed by the evidence provided here.

Replies to the comments of the reviewers

We thank the reviewers for their evaluation and comments, as well as their overall positive feedback. Below we provide a step-by-step reply to these comments, which are marked in blue. At the end of this document we include a list of changes that are made to the manuscript. These are marked in red in the revised manuscript.

Replies to the comments of reviewer #1

This is a very nice paper showing unambiguously that colloidal gels made of rough particles have significantly superior non-linear mechanical properties as compared to conventional gels made of “smooth” particles. This is something that people in the field knew (and often claimed) “intuitively”, at the qualitative level. The great contribution of this manuscript is to show it in a clean, quantitative way, using exactly the same kind of particles and modifying their roughness in a well-controlled way.

As such, the manuscript goes well beyond the existing literature and will certainly have a significant impact. Furthermore, the authors present two examples of applications where the enhanced mechanical properties of their gels could be key: extrusion printing and the fabrication of xerogels. For all these reasons, I strongly support publication in Nat. Comm.

We thank the reviewer for the positive assessment of the manuscript. We appreciate the comments below, which we believe we have all addressed and which helped us strengthen the manuscript.

I list below (in no particular order) several points I'd like the authors to address before final acceptance.

1) No real particle is actually “smooth”. Did the authors characterize the surface of the “smooth” silica particles? If not, they should cite previous works where this characterization was done for similar particles.

We agree with this comment and have added the RMS value for the smooth and rough particles, determined with AFM. The measured RMS values (RMS (smooth) = 0.7 nm, RMS (rough)=7.2nm) have been added to the text (highlighted in red, page 2 l.109-110).

2) The authors claim that the yield strain of colloidal gels is typically < 1% (abstract and line 45). I found this value quite small: some gels do have yield strains as large as 30% or more (see e.g. T. Gisler and D. A. Weitz, Strain Hardening of Fractal Colloidal Gels, Phys. Rev. Lett. 82, 1064 (1999)). Please revise this statement.

The specific publication refers to strain hardening suspensions which are only found under very specific conditions (density matched, low volume fractions). To accommodate the reviewers' comment we have revised the statement to be more precise and quote the arguments from the textbook of Mewis and Wagner (Colloidal suspension rheology Cambridge university press, 2012, chapter 6, p.209). who also state: “The yield strain is often very small, on the order of 1% or less, for gelled colloidal suspensions” which is congruent with our statement (we did not state all gels are, but typically)

The limiting strain is related to the deformation needed to move a bonded particle relative to a reference particle from the minimum separation distance in the bonded state (minimum of the potential, h_0) to just beyond the point of maximum force (h_y). The strain should scale as:

$$\uparrow \gamma = \frac{h_y - h_0}{2a}$$

With many potentials $h_y \approx 2h_0$ and this is in the nanometer range, a is the particle radius which is typically of 100 nm or above, so this yield strain can be of order 1% or below

3) Figure 1c: I find this sketch not very clear, one doesn't see very well the cluster densification

We thank the reviewer for this comment, this has been changed (figure 1 c. highlighted in red).

4) In all figures: increase the font size of the axis labels

We have made the plots consistent with the guidelines for authors.

5) Figures 2d, 2e of the main text and Figs. 3a, 3b of the SI: I suggest using the same color coding as in the rest of the paper, e.g. put each graph in a box, blue or red for smooth and rough particles, respectively.

Thank you, this has been incorporated.

6) Figures 2d, 2e: I suggest adding to the data label the % of the applied stress as compared to the plateau modulus (and quoting the value of the plateau modulus for each gel in the caption).

We thank the Reviewer 1 for their interesting input, we added the value of the corresponding plateau moduli to the caption and added the percentage of the applied stress where the sample fluidifies over the plateau modulus in the text to keep the figure as simple as possible (highlighted in red, page 5 caption figure 2).

7) Line 119: Fig. SI 3 shows stress sweeps, not strain sweeps.

Thank you, this has been corrected (highlighted in red, page 4 l.121).

8) Line 129: I suppose that the last point in the LVE is defined as the last point before G' drops by 10% or more upon further increasing the stress (10% with respect to what? The value of the previous point? The average of all previous points?). Please consider rephrasing this sentence to make it clearer.

We agree with Reviewer 1, that the sentence could be rephrased, therefore the wording has been adapted to:

«When considering the apparent oscillatory yield stress (Figure 2c.), which we define here as the last point in the LVE region, i.e., corresponding to a > 10 % difference in G' with respect to the average of the previous points (sometimes called the perturbative yield strain (cite Mewis and Wagner book), the rough particle systems yield at significantly higher stress amplitudes compared to the smooth particle systems at the same volume fraction (highlighted in red, page 4 l.121).

9) The abstract and introduction put emphasis on the small yield strain that is typically found

in colloidal gels, while the paper discusses almost only yield stress values. I understand that a stress-controlled rheometer was used, but it would be interesting to show (and discuss) at least the strain at yielding, e.g. as an additional panel after Fig. 2c of the main text. In the current version, strain at yielding is only briefly mentioned when discussing Table II.

We thank Reviewer 1 for their comment and added a figure representing the oscillatory yield strain for rough and smooth particle gels for different volume fractions in figure SI Figure 3 c. and refer to the panel in the text (highlighted in red, p.4 l.122).

10) Table II: it would be useful to mention also the thickness of the stabilizing layer, both at low and high temperatures.

We agree with Reviewer 1 comment and added the thickness of the stabilizing layer that was measured with AFM (highlighted in red p.8 l.185-186).

11) Figures 3b-e and 3g-j: it would be nice to add some quantitative characterization of the length scale of the heterogeneities seen in the images, rather than discussing them just at the qualitative level. Perhaps a straightforward Fourier transform of the images could be informative, could the authors give it a try?

We thank Reviewer 1 for their feedback and performed Moran's I analysis of the images, which is represented in SI Figure 5 o and explained in SI section 2 (highlighted in red).

12) Please define the acronym BET, and provide details on the BET measurements in the Methods section.

Thank you, this has been corrected (highlighted in red p.9 l.215).

13) Consider renaming the label "II. Aerogels" of Fig. 4 to "II. Xerogels" for consistency with the rest of the text and figure captions.

Thank you, this has been corrected.

14) I'm puzzled by the modest increase of the surface/mass ratio in gels as compared to powders, for both smooth and rough particles (Fig. 4l). Did the authors try to produce xerogels starting with gels at ϕ lower than 25%? I understand that the part on xerogels is, at this stage, just a proof-of-concept and that improving the BET ratio significantly would probably require a lot of work. Nevertheless, it would be nice to show data where the xerogel has properties that differ more markedly from those of powders, and more markedly between smooth and rough particles (the relative increment of the BET signal from the powder to the gel is in fact very similar for both kinds of particles)

Indeed, the increase of the surface area is quite modest in comparison with xerogels in literature. However, the particles used here are not engineered for the purpose of making xerogels, in terms of size and surface coating. First, the primary particles used in this work are an order of magnitude larger than traditionally used xerogel building blocks. Furthermore, the thermoreversible nature of the gels make the xerogel fabrication quite tricky because handling of the gels are room temperature yields to their fluidification.

Therefore, the relatively high volume fraction of 25 v% was used for the purpose of proof-of-concept to compare the xerogel fabrication using rough and smooth building blocks.

15) Ref 29 needs to be fixed

Thank you, this has been corrected.

Replies to the comments of reviewer #2

The authors create thermo-reversible particle based gels and perform experiments to distinguish their behavior from smooth particles. I do find the work to be compelling, interesting and novel, and worthy of publication, but I require the authors to make some major/minor improvements.

Major comments:

- The data shown in the main text lacks any form of statistical analysis. For example, in table 1, where is the average critical yield strain? Simply put. I am sure the authors performed many, if not hundreds of experiments, and yet none of this averaging is present in the manuscript. The reader is left to conclude that individual experiments are performed. The authors are required to add some statistical analysis.

We thank with Reviewer 2 for their comment, clarified the text and we added statistics on plot 2b. based on the standard deviation of the LVE region of the plateau modulus. Further, in the SI Figure 3 addresses the issue of reproducibility of the moduli after cycling the temperature and we added a comment in the text.

- Throughout the manuscript, the authors discuss the individual particle behavior and an 'interlocking'. While this is likely true, there is no evidence in this paper to confirm this. The authors should either indicate that this is logical speculation, or show clear individual particle analyses to confirm their hypothesis - it is a hypothesis. Particle based simulations would be one logical route to gain this level of detail or in-situ shear confocal microscopy. Both however are likely beyond the scope of this work.

We thank with Reviewer 2 for their comment; however, we would argue that by changing the size of the asperity particles (see table 1) and obtaining a gradual increase in critical strain as the asperity size increases, this is a clear indication on the underlying interlocking mechanism. Currently, we are also performing additional experiments using a colloidal probe AFM to elucidate the nature of the interparticle contacts as a function of temperature and the data indicate the presence of interlocking of asperities between particles. As mentioned by the Reviewer, these data are also beyond the scope of this work and will be published separately, but they contribute to strengthen our statement.

Minor Comments:

Page 3: by far the most common colloidal gels are made with smoothly droplets, i.e. emulsion. why has a discussion of these gels been ignored here?

We did not include emulsions or attractive emulsions in the discussion, as the focus lies on solid particle gels. However, Pickering-Ramsden emulsions are a parallel that we can draw when comparing rough and smooth primary building blocks.

Page 3: again, why no statistics on yielding? A thermos-reversible system allows for this quite easily.

We have this information in figure 3 of the SI and added comments to the text.

Page 6: The authors state: "In a sparsely populated system, where particles have space to rearrange, surface roughness may bring additional stability for network formation, while at higher volume fractions ($\phi > 0.1$), the system is more crowded, meaning that there is no need for an additional interlocking mechanism to contribute to network formation." I find this

statement to be unsupported by the evidence. it is quite a bold claim and given the highest volume fraction of 0.30 which is not that 'crowded' there is still amply local free volume (nearest neighborhood) for particles to rearrange via rolling/sliding. the authors should restate or better verify this statement.

We agree with the comment and suggest to clarify the statements as follows:

«In a sparsely populated system close to the percolation threshold, the additional surface area of the rough particles allows for stronger network formation, while at higher volume fractions ($\phi > 0.1$), the system is more crowded, meaning that there is no need for additional surface area to contribute to network formation.»

(highlighted in red, p.6, l.126-129)

Page 6: The authors state “The rough particle gel starts to fluidify when it is sheared at 180 Pa for 100 s (corresponding to a strain of 18.9 %), showing a very sudden and dramatic increase in strain which was also observed in the oscillatory measurements described above. “ this creep data is quite striking (and also lacking in statistics) and should be discussed more than this limited discussion. For instance, is there an origin to the abrupt yielding behavior in the rough particles? Were proper yielding statistics performed? How does this compare with past creep yield experiment on colloidal gels, in particular J. Sprakel, et. al. PRL 2011?

We suggest clarifying the statements as follows:

«The rough particle gel starts to fluidify when it is sheared at 180 Pa for 100 s (corresponding to a strain of 18.9 %), showing a very sudden and dramatic increase in strain which was also observed in the oscillatory measurements described above. We hypothesize, that this sudden fluidification is due to a general breaking of many interparticle bonds at the same time for the rough particles, whereas smooth particles tend to rearrange as they are being sheared.» (highlighted in red, p.6, l.140-145)

We added the reference to the Sprakel work with respect to the catastrophic failure, however in these experiments the critical strain remains very small, and the rupture is attributed to the presence of mesoscopic structures as inferred from two distinct exponential regimes with applied stress. This difference is not observed here (for the rough systems) and we would like to refrain from speculation about their existence as the larger yield strains in our rough systems suggest a more homogeneous microstructure. We have added a comment in the text to accommodate the reviewers' concern but would like to refrain from speculation.

Page 7 : The authors state “The smooth particle gel never recovers to the initial plateau modulus (indicated by the continuous horizontal line), even for much longer times than 5 min.” What is meant by 'initial plateau modulus'? Is this the modulus formed after the dispersion has been cooled in the rheometer? The shear history here is very important to mention compared to typical non-thermoreversible gels which have inherent loading history. This strengthens the authors' conclusions.

We thank Reviewer 2 for this comment and would like to specify that the initial plateau modulus is the one indicated by the straight line, which is the modulus measured after forming a gel through cooling of a particle suspension. Therefore, this gel is not biased through any shear history before the measurement. The shear history effect is show through the recovery after shearing the gel at 100 % strain for 1 min, as indicated in the method

section. We thank the reviewer for pointing out, and hope this clarification strengthens the conclusion.

Page 7: the authors state “The dense clusters that occur during the shearing of smooth particle gels are not able to separate when shear subsides, and thus the gel will reform with much larger and more compact primary clusters.” While this statement is probably correct, the confocal images do not prove this. When considering, panel b compared to e, it is very difficult to conclude that denser clusters are formed. No statistical analysis of the structure factor is performed and should be. Indeed, the best and most reliable measurement would be SAXS determination, but I do understand that this is too much experimentation. I suggest the author either perform some analyses of their confocal images, or tone down the strength of their statement as it is currently speculation and not obvious from the images provided - images being very subjective and also highly subjected to the bias of the writer.

We thank Reviewer 2 for their comment, we have attempted SAXS experiments, which were difficult to realize due to difficulty of temperature control and the size of the particles. SALS experiments would be subject to too much multiple scattering to enable quantitative analysis. FFT analysis of the confocal images were inconclusive, as there are several lengthscales present in the colloidal gel. In order to still have a quantitative assessment of the confocal images, we performed a Moran’s I analysis which is detailed in the SI and the results are represented in figure S5 o. (highlighted in red).

Page 7: the authors state “In the confocal images of the rough particle gel (Figure 3 b.), the network structure prevails throughout all images, showing that interlocking of the surface asperities hinders flow densification.” This conclusion is again speculation and should be mention as such. And the figure reference is incorrect, it should not be ‘b’.

We thank Reviewer 2 for their comment, we adjusted the figure reference (highlighted in red, p.8, l.169), concerning the wording, we would like to answer similarly as to comment #2.

Page 8: the authors state “It is apparent from the images, that the printing quality of the rough particle gel is superior to the smooth particle gel, primarily because of the previously described recovery ability of the rough particle.” What is apparent? Please describe better the print quality. This sentence is too vague and non-descriptive. Additionally, the recovery is 0.5minutes for the rough particle gel, as shown in the above figure. This is still extremely slow recovery compared to millisecond for polymeric 3D printing. The authors must discuss the printing velocity used, and how this affected the print quality. The methods section is severely lacking in detail for this application.

We thank Reviewer 2 for this comment. We suggest expanding the sentence to:

«It is apparent from the images that the printing quality of the rough particle gel is superior to the smooth particle gel, as the printed “filament” across the different layers clearly retain a more defined shape for the rough particle system.» (highlighted in red, p8, l.196-198).

Furthermore, we have specified the printing speed of 1 mm/s in the method section, which shows that there is enough time for a layer to fully recover before the next layer is printed on top.

Conclusion: Again there is strong speculation that 'interlocking' is the origin of the rheological difference, but no local individual particle data is provided in this manuscript. I do not believe the absence is too impactful of the results, rather than the authors temper their language to indicate that it is hypothesized and not confirmed by the evidence provided here.

Thank you, we have adapted the manuscript accordingly.

REVIEWERS' COMMENTS

Reviewer #1 (Remarks to the Author):

The authors have convincingly addressed all my remarks: I'm happy to recommend their manuscript for publication in Nature Comm.

Reviewer #2 (Remarks to the Author):

The authors have address all of my concerns well and also the concerns of the other reviewer. In addition, I found the calculation of Moran's I to be quite an inspired choice and helps clarify their conclusions elegantly without the need for more complex analyses.